

# intansv: an R package for integrative analysis of structural variations

Lihua Jia[1,2,*], Na Liu[1,*], Fangfang Huang[1], Zhengfu Zhou[3], Xin He[1], Haoran Li[1], Zhizhan Wang[1] and Wen Yao[1,4]

[1] National Key Laboratory of Wheat and Maize Crop Science, College of Life Sciences, Henan Agricultural University, Zhengzhou, Henan, China
[2] National Key Laboratory of Wheat and Maize Crop Science, College of Agronomy, Henan Agricultural University, Zhengzhou, Henan, China
[3] Wheat Research Institute, Henan Academy of Agricultural Sciences, Zhengzhou, Henan, China
[4] National Key Laboratory of Crop Genetic Improvement, National Center of Plant Gene Research (Wuhan), Huazhong Agricultural University, Wuhan, Hubei, China
[*] These authors contributed equally to this work.

## ABSTRACT

Identification of structural variations between individuals is very important for the understanding of phenotype variations and diseases. Despite the existence of dozens of programs for prediction of structural variations, none of them is the golden standard in this field and the results of multiple programs were usually integrated to get more reliable predictions. Annotation and visualization of structural variations are important for the understanding of their functions. However, no program provides these functions currently as far as we are concerned. We report an R package, intansv, which can integrate the predictions of multiple programs as well as annotate and visualize structural variations. The source code and the help manual of intansv is freely available at https://github.com/venyao/intansv and http://www.bioconductor.org/packages/devel/bioc/html/intansv.html.

## INTRODUCTION

Next-generation sequencing (NGS) has greatly enhanced our ability to detect genomic variations between individuals which is the key to the understanding of phenotype variations and diseases. Although the third-generation sequencing technologies have emerged as new options to detect genomic variations, NGS is still the preferred approach concerning the sequencing cost and accuracy (*Spealman, Burrell & Gresham, 2019*). Genomic variation is comprised of single nucleotide polymorphisms (SNPs), insertions/deletions (indels) and structural variations (SVs). SNPs have been widely detected and utilized in quantitative trait locus (QTL) mapping and genome wide association study (GWAS) while SVs, which usually cause large effects, are more difficult to be detected (*Alkan, Coe & Eichler, 2011*; *Sadowski et al., 2019*).

Currently, many programs have been developed to detect SVs based on NGS (*Abyzov et al., 2011*; *Becker et al., 2018*; *Chen et al., 2009*; *Layer et al., 2014*; *Rausch et al., 2012*).

Corresponding author
Wen Yao, yaowen@henau.edu.cn

The algorithms employed by these programs could be categorized into three groups: (1) read-pair methods which analyze the span size and mapping orientation of paired-end reads and their inconsistency from expectation (*Chen et al., 2009*), (2) split-read algorithms which align the unmappable end of read pairs of which only one end can be mapped to the reference genome using the split-read alignment (*Ye et al., 2009*) and (3) read depth analysis which utilizes the increase and decrease in sequence coverage (*Abyzov et al., 2011*). However, the results of different tools have been reported to be discordant with each other and none of the currently existing programs is the golden standard in this field (*Lin et al., 2015*). As a result, the outputs of different tools were usually integrated to get more reliable predictions (*Sudmant et al., 2015*).

SVMerge (*Wong et al., 2010*), HugeSeq (*Lam et al., 2012*) and iSVP (*Mimori et al., 2013*) are tools to detect SVs by integrating different SV callers as part of their pipelines. BreakDancer (*Chen et al., 2009*), Pindel (*Ye et al., 2009*), SECluster, RDXplorer (*Yoon et al., 2009*) and RetroSeq (*Keane, Wong & Adams, 2012*) are integrated in the pipeline of SVMerge. HugeSeq and iSVP only allow the use of supplied default callers. HugeSeq integrates four SV callers including BreakDancer, Pindel, CNVnator (*Abyzov et al., 2011*) and BreakSeq (*Lam et al., 2010*) while iSVP incorporates BreakDancer and Pindel as part of its pipeline. The installation and configuration of these tools are challenging for users with no computing experience as different SV callers are implemented with varying programming languages and are glued together by these pipelines utilizing Linux shell, Perl and Python. To use these pipelines, the users must successfully install several dependency software required by these pipelines in advance. In addition, it is important to annotate SVs and visualize the distribution of SVs in the whole genome or a specified genomic region. However, no software provides these functions simultaneously for now.

We report intansv, an R package which is easy to install and use, aiming to integrate the output of multiple popular programs, annotate the effects of SVs to genes and visualize the SVs in the whole genome or specified genomic regions (Fig. 1). The installation of intansv is independent to the installation of SV callers as it only relies on the output of SV callers. The intansv package supports the integration of the results of some or all seven prevalent SV callers.

## MATERIAL AND METHODS

The NGS data of Zhenshan 97 were downloaded from NCBI SRA (accession number SRA012177) and were aligned to the rice Nipponbare reference genome (MSU version 7.0) using BWA (version 0.6.1-r104) with default parameters (*Kawahara et al., 2013*; *Li & Durbin, 2009*). The alignment result was processed by SAMtools and stored in BAM format (*Li et al., 2009*). The alignment result in BAM format and the reference genome sequence are essential inputs for SV callers. Some of the SV callers use only the alignment result in BAM format as the input file while some of the SV callers also need the reference genome sequence file as the input file. All the seven SV callers were run with default parameters.

For each SV caller, various information was output to evaluate the quality of each predicted SV, including read mapping quality, number of discordant read pairs supporting
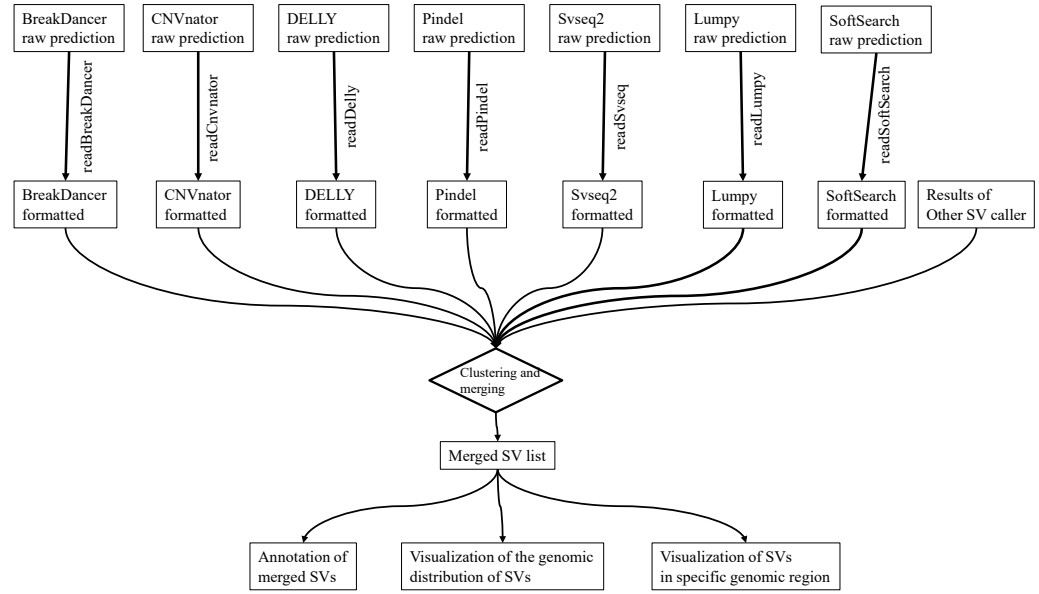

**Figure 1** **The pipeline of intansv for integrative analysis of structural variations (SV).** The raw outputs of SV callers are first processed by intansv to get user-friendly results. During this process, the low-quality predictions were filtered out. The processed outputs of different SV callers are then merged for downstream analysis, including annotation and visualization.

the SV, number of split reads supporting the SV, etc. The information was utilized to filter low quality predictions of these programs. For example, SVs predicted by BreakDancer with fewer than 3 discordant read pairs support were filtered by intansv by default. Each SV was assigned a score representing the quality of the SV by BreakDancer and SVs with a score smaller than 60 were filtered by intansv. In addition, SVs shorter than 100 bp or longer than 10 Mb predicted by any SV caller were also filtered by intansv by default. For the results of Pindel, SVs supported by fewer than 3 split reads were filtered. For the results of SoftSearch, SVs supported by fewer than 3 split reads or supported by fewer than 3 discordant read pairs were filtered (*Hart et al., 2013*).

To cluster SVs predicted by different SV callers, we first combined the same type of SVs of different SV callers and detected the coordinate overlapping between SVs of the same type. The findOverlaps function of the R package GenomicRanges was used to detect the coordinate overlapping between different SVs, which could be represented as genomic ranges (*Lawrence et al., 2013*). Then the coordinate overlapping percentage between pairwise overlapped SVs was calculated. We further define the distance between pair-wise SVs as 0 if they have a reciprocal coordinate overlap of more than 80%. The distance between non-overlapped pair-wise SVs or pair-wise SVs with a reciprocal coordinate overlap of no more than 80% was defined as 1. In this way, a distance matrix could be constructed for combined SVs of the same type. We can then use hierarchical clustering to divide the combined SVs of the same type into different clusters. A cluster of SVs supported by two or more methods were then merged as a unified SV by taking the mean of the start coordinates of all SVs in the cluster as the start coordinate of the merged SV and taking the mean of
**Table 1** Structural variations reported by different programs using NGS data of a rice variety (Zhenshan 97).

| Methods | Deletion | Duplication | Inversion |
|---|---|---|---|
| BreakDancer raw predictions | 9010 | NA[a] | 1202 |
| BreakDancer formatted predictions | 6890 | NA | 153 |
| Pindel raw predictions | 7641 | 3672 | 1724 |
| Pindel formatted predictions | 4322 | 561 | 1180 |
| CNVnator raw predictions | 2806 | 566 | NA |
| CNVnator formatted predictions | 2805 | 566 | NA |
| Svseq raw predictions | 7694 | NA | NA |
| Svseq formatted predictions | 1556 | NA | NA |
| DELLY raw predictions | 9034 | 497 | 1746 |
| DELLY formatted predictions | 5881 | 214 | 219 |
| Lumpy raw predictions | 7439 | 289 | 5 |
| Lumpy formatted predictions | 4684 | 109 | 3 |
| SoftSearch raw predictions | 580 | 14 | 86 |
| SoftSearch formatted predictions | 263 | 4 | 16 |
| intansv | 6255 | 139 | 92 |

Notes.

[a] NA, this program doesn't support prediction of this type of SV.

the end coordinates of all SVs in the cluster as the end coordinate of the merged SV. Any cluster of SVs supported by only one method were discarded.

# RESULTS

The intansv package takes the predictions of multiple SV callers as input and outputs an integrated list of SVs. To use intansv, the users need to run SV callers and prepare the output files as input to intansv. Here, we use the NGS data of a rice variety Zhenshan 97 to illustrate the versatility of intansv (Materials and Methods) (*Xie et al., 2010*). Seven different programs, BreakDancer (*Chen et al., 2009*), CNVnator (*Abyzov et al., 2011*), DELLY (*Rausch et al., 2012*), Pindel (*Ye et al., 2009*), SVseq2 (*Zhang, Wang & Wu, 2012*), Lumpy (*Layer et al., 2014*) and SoftSearch (*Hart et al., 2013*) were used with default parameters to predict SVs based on the alignment of the NGS data of Zhenshan 97 to the Nipponbare reference genome (Table 1). The Nipponbare reference genome is composed of 374.5 million nucleotide base pairs with a GC content of 43.5%. The NGS data of Zhenshan 97 provided over 14× coverage of the Nipponbare reference genome.

The first step to use intansv is the reading of the output of SV callers. The intansv package provides reusable and efficient tools to deal with the outputs of the seven programs as they are tedious and not user-friendly for downstream analysis (Fig. 1). During the reading process, low quality predictions given by these programs were filtered out by intansv (Table 1, Materials and Methods). In addition, overlapped predictions output by a single SV caller would be resolved by intansv.

BreakDancer is able to predict deletions and inversions. The prediction of all type of SVs were provided as a single file by BreakDancer. The output of BreakDancer contains

9010 deletions and 1202 inversions in the whole genome of Zhenshan 97 (Table 1). The following R script shows the processing of the output of BreakDancer by intansv. The parameter breakdancer.file.path stores the file path of the output file of BreakDancer in the disk. A total of 6890 deletions and 153 inversions in the whole genome of Zhenshan 97 were read into R as an R list by intansv after filtering low quality predictions (1034 deletions and 24 inversions for chromosome chr05 and chr10 in this demo). The evaluated score and evidence supporting each SV were also read into R. Each element of the R list is an R data frame containing the results of a type of SV, which can be easily wrote into the disk as text files using the write.table function of R.

>breakdancer <- readBreakDancer(breakdancer.file.path)
>str(breakdancer)
List of 2
$ del:'data.frame': 1034 obs. of 5 variables:
..$ chromosome: chr [1:1034] "chr05" "chr05" "chr05" "chr05" ...
..$ pos1: int [1:1034] 65575 86452 120264 153666 201845 208214 230521 276048 ...
..$ pos2: int [1:1034] 65949 87419 127693 153829 201959 208577 230654 276182 ...
..$ size: num [1:1034] 353 988 7382 107 114 ...
..$ info: chr [1:1034] "score=88;PE=5" "score=99;PE=20" "score=83;PE=6" ...
$ inv:'data.frame': 24 obs. of 5 variables:
..$ chromosome: chr [1:24] "chr05" "chr05" "chr05" "chr05" ...
..$ pos1: int [1:24] 1291603 6942362 12014477 18770973 20583909 29431819 ...
..$ pos2: int [1:24] 1291649 6944726 12016019 18771402 20584709 29432511 ...
..$ size: num [1:24] -105 2186 1334 452 320 ...
..$ info: chr [1:24] "score=99;PE=3" "score=99;PE=5" "score=99;PE=6" ...
- attr(*, "method")= chr "BreakDancer"

CNVnator is able to predict deletions and duplications. The final output of CNVnator usually contains several files and each file is the output for a single chromosome. All these files should be put in the same directory and the path of this directory should be input to the function readCnvnator of intansv. Then the output of CNVnator will be read into R. The directory given to readCnvnator should only contain the final output files of CNVnator. The output of CNVnator contains 2,806 deletions and 566 duplications in the whole genome of Zhenshan 97 (Table 1). The following R script shows the processing of the output of CNVnator by intansv. The parameter cnvnator.dir.path stores the directory path of all the output files of CNVnator in the disk. A total of 2,805 deletions and 566 duplications in the whole genome of Zhenshan 97 were read into R (369 deletions and 113 duplications for chromosome chr05 and chr10 in this demo).

>cnvnator <- readCnvnator(cnvnator.dir.path)
>str(cnvnator)
List of 2
$ del:'data.frame': 369 obs. of 5 variables:
..$ chromosome: chr [1:369] "chr05" "chr05" "chr05" "chr05" ...
..$ pos1: num [1:369] 104101 230401 348301 1089301 1524301 ...
..$ pos2: num [1:369] 110700 248400 350100 1103100 1529700 ...

```
..$ size: num [1:369] 6599 17999 1799 13799 5399 ...
..$ info: chr [1:369]
"eval1=0.0100681;eval2=1386200000;eval3=10.1189;eval4=1690690000" ...
$ dup:'data.frame': 113 obs. of 5 variables:
..$ chromosome: chr [1:113] "chr05" "chr05" "chr05" "chr05" ...
..$ pos1: num [1:113] 405001 973201 1346401 5036101 6419101 ...
..$ pos2: num [1:113] 408900 977700 1359000 5055600 6423300 ...
..$ size: num [1:113] 3899 4499 12599 19499 4199 ...
..$ info: chr [1:113] "eval1=0.0169421;eval2=0;eval3=0.00545604;eval4=0" ...
- attr(*, "method")= chr "CNVnator"
```

As SVs are much more complex to detect than single nucleotide variations, predictions of different programs were usually integrated to get more reliable results. The second step to use intansv is to integrate the predictions of multiple SV callers. The intansv package provides efficient functions to cluster and merge the predictions of different programs (Materials and Methods). Predictions of varying programs were first clustered together if they have reciprocal coordinate overlap of more than a specified percentage (default value 80%) (Fig. 2). Then clusters supported by more than a predefined number (default value 2) of methods were merged as one SV respectively while the rest clusters were discarded (Fig. 2). The intansv package is also able to integrate the predictions of other SV callers with appropriate file format provided. By integrating the results of different SV callers, intansv provides the most reliable and comprehensive SV predictions (Table 1). The following R script shows the integration of the output of BreakDancer, Pindel, CNVnator, DELLY, SVseq2 by intansv. The integrated results contain 851 deletions, 13 duplications and 16 inversions for chromosome chr05 and chr10 of the Zhenshan 97 genome. A total of 6,255 deletions, 139 duplications and 92 inversions in the whole genome of Zhenshan 97 were detected by integrating the results of seven SV callers (Tables 1, 2). The average length of the 6,255 deletions, 139 duplications and 92 inversions were 3,817 bp (ranging from 103 bp to 312,652 bp), 4,0174 bp (ranging from 109 bp to 898,618 bp) and 92,364 bp (ranging from 143 bp to 904,615 bp), respectively (Figs. 3A–3C). The distribution of SVs in different chromosomes is not even (Fig. 3D).

```
>sv_all_methods <- methodsMerge(breakdancer, pindel, cnvnator, delly, svseq)
>str(sv_all_methods)
List of 3
$ del:'data.frame': 851 obs. of 5 variables:
..$ chromosome: chr [1:851] "chr05" "chr05" "chr05" "chr05" ...
..$ pos1: num [1:851] 65576 86449 120352 208217 280413 ...
..$ pos2: num [1:851] 65949 87419 127768 208564 281198 ...
..$ methods: chr [1:851] "BreakDancer:Delly" "BreakDancer:Pindel:Delly" ...
..$ info: chr [1:851] "score=88;PE=5:SU=5" "score=99;PE=20:SR=11;score=42:SU=21" ...
$ dup:'data.frame': 13 obs. of 5 variables:
..$ chromosome: chr [1:13] "chr05" "chr05" "chr05" "chr05" ...
..$ pos1: num [1:13] 120449 910600 1615000 3087200 5696556 ...
..$ pos2: num [1:13] 127870 911326 1616028 3090903 5699328 ...
```
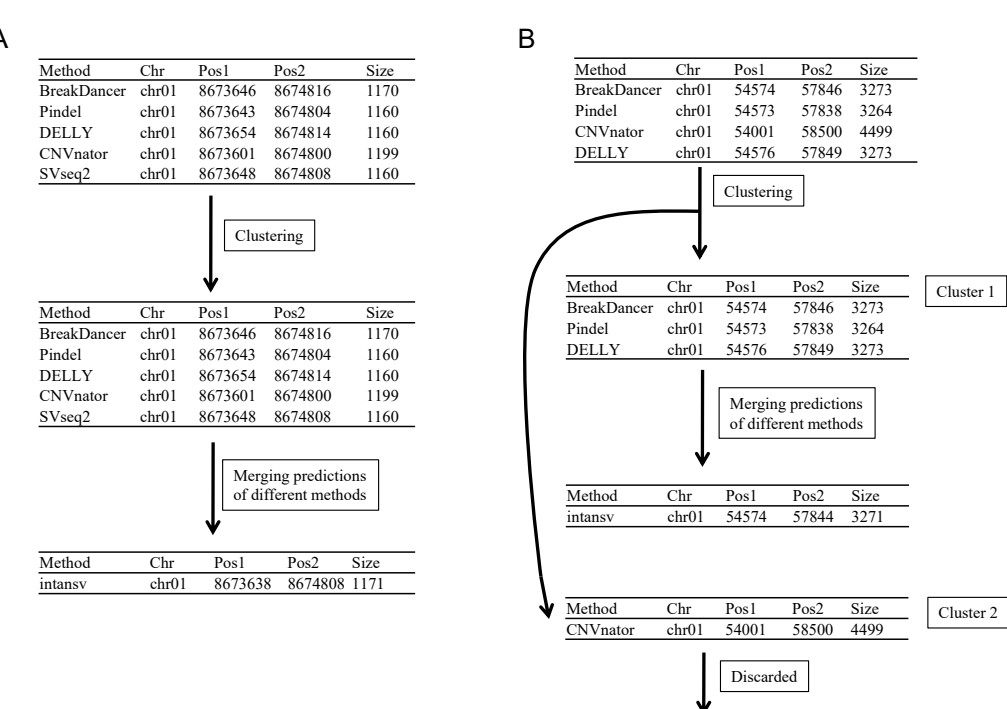

**Figure 2** **Merging the predictions of different SV callers.** Predictions of different programs were first clustered together if they have reciprocal coordinate overlap of more than a defined threshold (default 80%). Then clusters supported by more than two methods were merged as one SV (A and Cluster 1 in B) respectively while the rest clusters supported by no more than two methods were discarded (Cluster 2 in B).

**Table 2** **Integrated prediction of SVs by intansv based on the NGS data of a rice variety (Zhenshan 97).**

| Chromosome | Deletion | Duplication | Inversion |
|---|---|---|---|
| chr01 | 717 | 12 | 10 |
| chr02 | 591 | 14 | 12 |
| chr03 | 521 | 5 | 6 |
| chr04 | 562 | 17 | 8 |
| chr05 | 448 | 18 | 5 |
| chr06 | 565 | 7 | 3 |
| chr07 | 590 | 13 | 11 |
| chr08 | 434 | 14 | 4 |
| chr09 | 307 | 6 | 4 |
| chr10 | 455 | 9 | 9 |
| chr11 | 588 | 13 | 14 |
| chr12 | 477 | 11 | 6 |

..\$ methods: chr [1:13] "Pindel:Delly" "Pindel:Delly" "Pindel:Delly" "Pindel:Delly" ...
..\$ info: chr [1:13] "SR=7;score=14:SU=6" "SR=3;score=4:SU=12" ...
\$ inv:'data.frame': 16 obs. of 5 variables:
..\$ chromosome: chr [1:16] "chr05" "chr05" "chr05" "chr05" ...

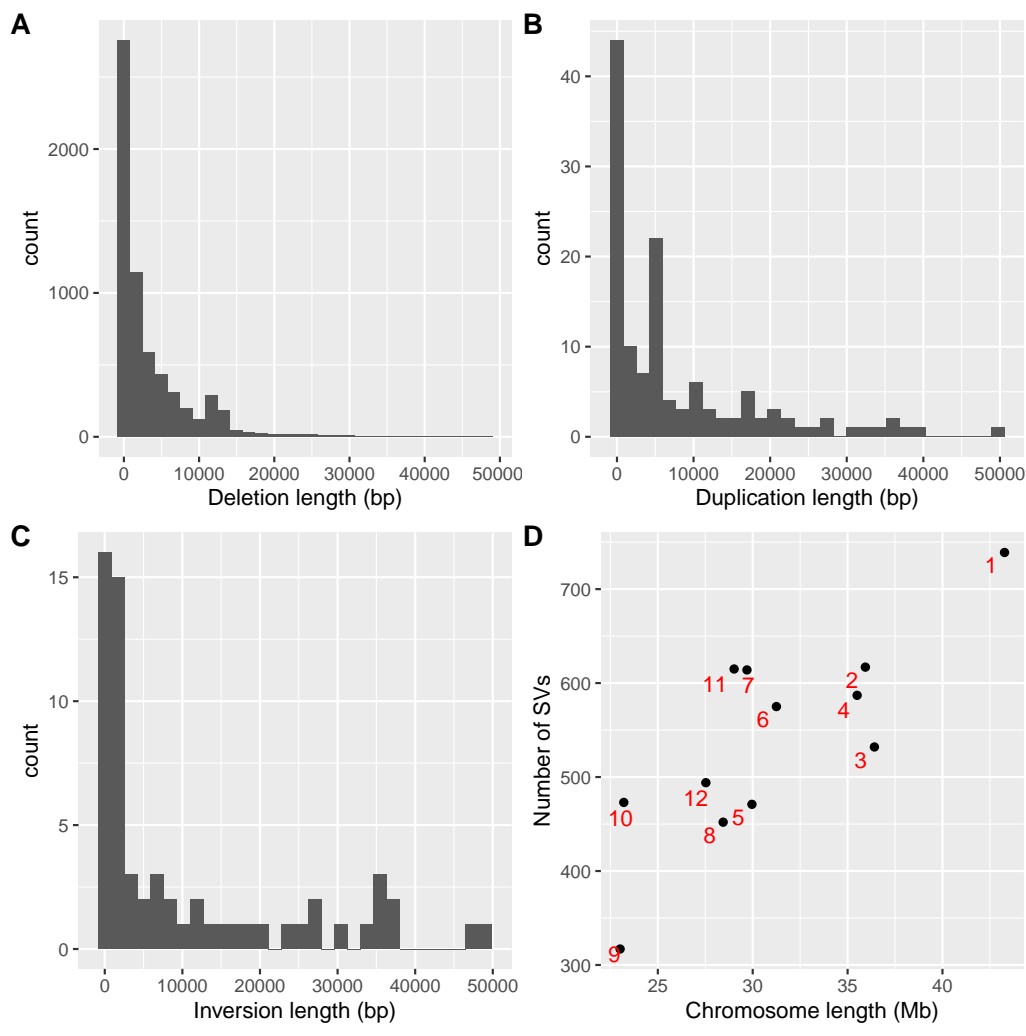

**Figure 3** **Lengths and genome distributions of SVs identified by intansv using the dataset of Zhenshan 97.** (A) Histogram of the lengths of deletions identified by intansv. (B) Histogram of the lengths of duplications identified by intansv. (C) Histogram of the lengths of inversions identified by intansv. (D) Comparison of the size of each chromosome with the number of SVs identified in each chromosome. Each chromosome is represented as a black point with the identifier labeled beside the corresponding point.

..$ pos1: num [1:16] 6372420 6942364 12014448 15089985 28357689 ...
..$ pos2: num [1:16] 6453749 6944728 12015959 15158230 28391740 ...
..$ methods: chr [1:16] "Pindel:Delly" "BreakDancer:Delly" ..
$ info: chr [1:16] "SR=11;score=12:SU=6" "score=99;PE=5:SU=5" ...

Analysis of the effect of SVs to the structure of genes is essential for the understanding of their functions. The third step to use intansv is to annotate the effect of SVs to the genome structure. For each identified SV, intansv provides details on the overlapping between this SV and genes or the elements of genes (Fig. 4A). Functions of the R package GenomicRanges was utilized in the intansv package to dissect the overlapping between SV and gene elements (*Lawrence et al., 2013*). Based on the NGS data of Zhenshan 97, we
A

| chromosome | pos1 | pos2 | methods | info | tag | start | end | strand | ID |
|---|---|---|---|---|---|---|---|---|---|
| chr05 | 65582 | 65944 | BreakDancer:DELLY | score=88;PE=5:PE=5;SR=0 | NA | NA | NA | NA | NA |
| chr05 | 86446 | 87423 | BreakDancer:Pindel:DELLY | score=99;PE=20:SR=11;score=42:PE=18;SR=0 | NA | NA | NA | NA | NA |
| chr05 | 120360 | 127760 | BreakDancer:Pindel:DELLY | score=83;PE=6:SR=4;score=5:PE=7;SR=0 | NA | NA | NA | NA | NA |
| chr05 | 208219 | 208562 | BreakDancer:DELLY | score=99;PE=9:PE=17;SR=0 | gene | 207853 | 209693 | - | LOC_Os05g01330 |
| chr05 | 208219 | 208562 | BreakDancer:DELLY | score=99;PE=9:PE=17;SR=0 | mRNA | 207853 | 209693 | - | LOC_Os05g01330.1 |

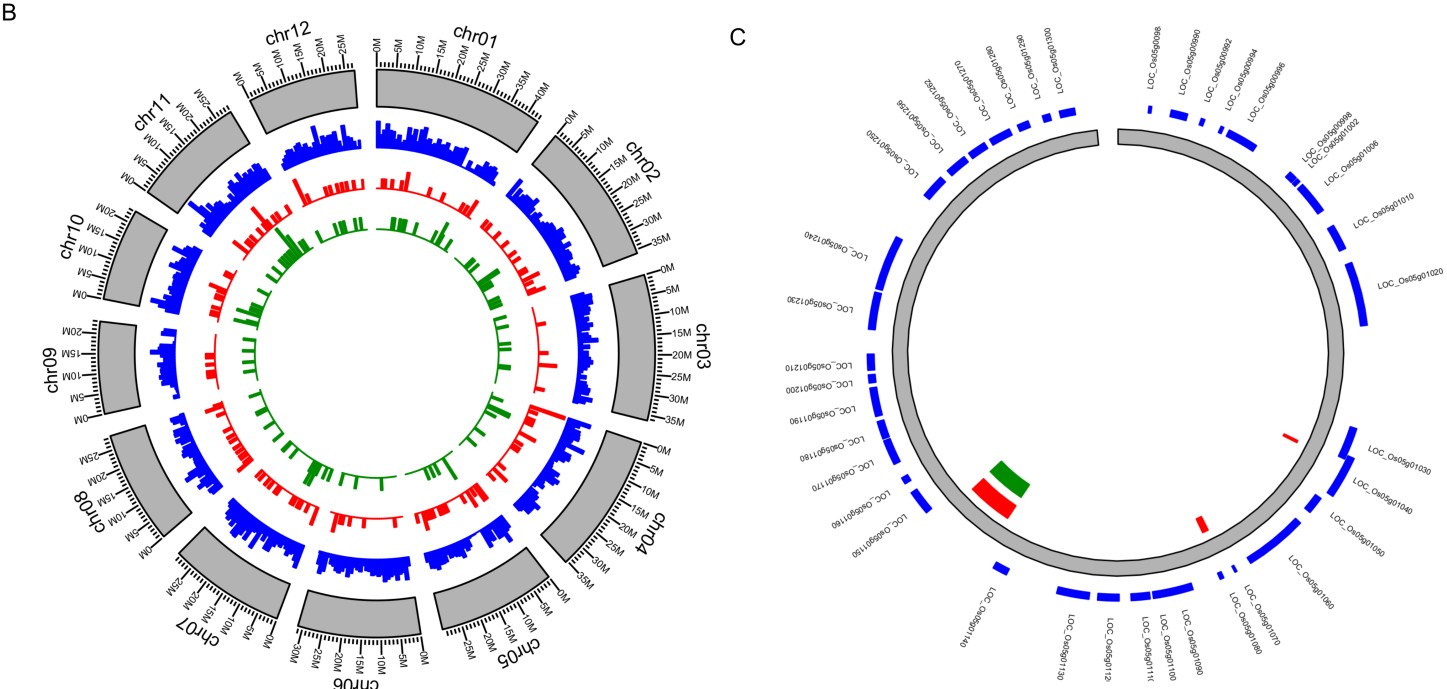

**Figure 4** **The demonstration of functions of intansv using the dataset of Zhenshan 97.** (A) The integrated output of intansv based on the predictions of seven programs including BreakDancer, Pindel, CNVnator, DELLY, Svseq2, Lumpy and SoftSearch. Chromosome, the chromosome ID of an SV. Pos1, the start coordinate of an SV. Pos2, the end coordinate of an SV. info, Information on an SV reported by different SV callers. (B) The distribution of SVs in the whole genome (blue, deletions; red, duplications; green, inversions). (C) The demonstration of SVs and genes in a specified genomic region (blue, genes; red, deletions; green, duplications).

identified 4,772 genes affected by deletions, 912 genes affected by duplications and 1,338 genes affected by inversions (Tables S1–S3).

The final step to use intansv is to visualize the distribution of SVs in the whole genome or a specified genomic region. The chromosomes were split into windows of specific size and the number of SVs within each window were counted and displayed as circular bar plot (Fig. 4B). For a specified region, the SVs and genes in it were displayed as circular rectangles to show the effects of SVs to the structures of genes and elements of genes (Fig. 4C). Functions of the R package ggbio were utilized to realize this functionality in intansv (*Yin, Cook & Lawrence, 2012*).

## DISCUSSION

We provide an R package, intansv, for integrative analysis of structural variations. It can process raw output of seven prevalent programs, merge predictions of different programs,

annotate effects caused by SVs to gene and its elements, visualize SVs distribution in the whole genome or a specified genomic region, and output an integrated SV list as a single file. Visualization of SVs in the whole genome is helpful for identifying SV hotspots while visualization of SVs in specific genomic regions is beneficial to uncover the potential effects of SVs through comparing SVs with genomic features or other experimental data. The intansv package is a program with incorporate functions to deal with miscellaneous output styles of different SV callers which has great potential to be widely accepted by the users. The parameter values of all the functions of intansv can be adjusted by the users to meet the criteria of different users. The source code and the help manual of intansv are deposited in GitHub (https://github.com/venyao/intansv). The functionalities of old versions of the intansv package and the example output file format of different SV callers integrated by intansv is provided at https://venyao.github.io/intansv/.

## CONCLUSION

The discovery and genotyping of SV is very important for understanding its role in phenotype variations. Precise and comprehensive detection of SV is still difficult due to the complexity of SVs. We developed the intansv package which is able to integrate the predictions of multiple SV callers. New features were implemented in the intansv package including the annotation and visualization of SVs. With the development of NGS and long read sequencing technology, more and more programs for the prediction of SVs would be developed. We will further enhance the functionalities of intansv in the future by implementing new functions to deal with the output of more programs to liberate researchers from tedious work of data cleansing.

### Funding

This research was supported by the research start-up fund of Henan Agricultural University (30500581), the Scientific and Technological Project of Henan Province (182102110278), the Project of Henan Provincial Department of Education (18A210017) and the open funds of the National Laboratory of Wheat Engineering, Key Laboratory of Wheat Biology and Genetic Breeding in Central Huang-huai Region, Ministry of Agriculture, Henan Key Laboratory of Wheat Biology. The funders had no role in study design, data collection and analysis, decision to publish, or preparation of the manuscript.

### Grant Disclosures

The following grant information was disclosed by the authors:
Henan Agricultural University: 30500581.
Scientific and Technological Project of Henan Province: 182102110278.
Project of Henan Provincial Department of Education: 18A210017.
National Laboratory of Wheat Engineering.
Key Laboratory of Wheat Biology and Genetic Breeding in Central Huang-huai Region.
Ministry of Agriculture, Henan Key Laboratory of Wheat Biology.

## Competing Interests

The authors declare there are no competing interests.

## Author Contributions

- Lihua Jia and Na Liu performed the experiments, analyzed the data, prepared figures and/or tables, authored or reviewed drafts of the paper, and approved the final draft.
- Fangfang Huang analyzed the data, prepared figures and/or tables, authored or reviewed drafts of the paper, and approved the final draft.
- Zhengfu Zhou analyzed the data, authored or reviewed drafts of the paper, and approved the final draft.
- Xin He, Haoran Li and Zhizhan Wang analyzed the data, prepared figures and/or tables, and approved the final draft.
- Wen Yao conceived and designed the experiments, analyzed the data, prepared figures and/or tables, authored or reviewed drafts of the paper, and approved the final draft.

## Data Availability

The source code and the help manual of intansv are available at GitHub: https://github.com/venyao/intansv.

## Supplemental Information

Supplemental information for this article can be found online at http://dx.doi.org/10.7717/peerj.8867#supplemental-information.

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
