# Peer review of "intansv: an R package for integrative analysis of structural variations"

_PeerJ, doi:10.7717/peerj.8867_

## Round 0.1 · original submission · Major Revisions

Dear Dr. Jia and colleagues:

Thanks for submitting your manuscript to PeerJ. I have now received three independent reviews of your work, and as you will see, the reviewers raised some concerns about the research. Despite this, these reviewers are optimistic about your work and the potential impact it will lend to research on analyzing genotypic variation. Thus, I encourage you to revise your manuscript, accordingly, taking into account all of the concerns raised by the reviewers.

While the concerns of the reviewers are relatively minor, this is a major revision to ensure that the original reviewers have a chance to evaluate your responses to their concerns.

I look forward to seeing your revision, and thanks again for submitting your work to PeerJ.

Good luck with your revision,

-joe

Reviewer 1 ·

Basic reporting

no comment

Experimental design

no comment

Validity of the findings

no comment

Additional comments

In the manuscipt titled ‘intansv: an R package for integrative analysis of structural variations’, the authors provided an R package of integrating structural variations (SVs) from multiple SV-genotyping programs. The versatile package can be used to process raw results from some SV-genotyping programs, to merge formatted results, to annotate SV’s’ effects, and to visualize the SVs’ distribution. Considering the complexity of SV-genotyping, the package is very useful for biologists or geneticists. The manuscript is generally well written and the supplied data is sufficient. I still have two minor comments which the authors should be concerned.
1. As a proof, the SV results were obtained based on the NGS data of Zhenshan 97 using the ‘intansv’ package. It is suggested that the authors should provide some description or statistics for the genotyped SVs of Zhenshan 97 in the ‘Results’ part.
2. ‘versatile’ should be ‘versatility’ (Line 90).

·

Basic reporting

The article is explained in clear and unambiguous English.

The figures and tables are properly labelled and referenced in the article.

Experimental design

In the "Materials and Methods" section, the method used to filter out the Low-quality predictions can be added.

The process is explained well with a code sample.

Validity of the findings

In the discussion section, the benefits from the visualization of the results can be explained. As it was mentioned as a key improvement in the abstract (please refer to lines 39 and 40).

Please rephrase line number 214 - 216 in the conclusion section explaining the improvements made in this article and the future scope of this package.

Additional comments

This article talks about the addition of a new package to the open-source R software for the Integrative analysis of structural variations. The source code and various methods available in the package are provided. The visualization generated in this package is helpful to understand the results.

Reviewer 3 ·

Basic reporting

Good introduction, provides good background.
Clear english... Some sentences uses "you". change those.
Discussion could be expanded.
Some figures (for instance figure 2) need to be higher quality. Some parts are blurry.

Experimental design

Interesting package. The difficulty of using multiple programmes to SV detection is very real, if this package simplifies this it is a huge benefit.

Methods could use more a lot more detail. How were SVs deemed overlapping in from different programme outputs?
How many individuals used?
size of genome? GC content?
A bit more detail is needed.
There should also be:
- How SVs from different programes clustered? merged? annotated etc?

It is hard to see what was actually done .....

Validity of the findings

Result section is lacking. Reads more like a methods section with some raw output here and there. Summarise findings etc. Remove raw output. Maybe put in supplemental data...


The use of “you” may not be the best way to present results. Try using for example “All files should be put in the same directory….”


Very unclear what the findings were? final results? run time? etc...

A lot of the results section should actually be in the methods section.

Additional comments

The methods section need expanding overall. very little detail on what was actually done. See comments in methods section and figures. Unclear how SVs from different platforms were merged, clustered, formatted?

Results section could benefit from being re-organised and detailed.

What does the final output look like? how were SVs formatted?

Discussion to be expanded to emphasise the novelty of a package that uses multiple programmes and outputs a single, standardised, final output.

---

## Round 0.2 · accepted · Accept

Dear Dr. Jia and colleagues:

Thanks for again revising your manuscript. I now believe that your manuscript is suitable for publication. Congratulations! I look forward to seeing this work in print, and I anticipate it being an important resource for research analyzing genotypic variation. Thanks again for choosing PeerJ to publish such important work.

Best,

-joe

Reviewer 1 ·

Basic reporting

no comment

Experimental design

no comment

Validity of the findings

no comment

Additional comments

The authors have addressed all of my concerns. In my opinion, the manuscript can be accepted for publication.

·

Basic reporting

The article is explained in clear and unambiguous English.
The figures and tables are properly labelled and referenced in the article.

Experimental design

The method to filter the low-quality predictions are described in detail.

Validity of the findings

The use of visualization is explained in the Conclusion part.